# Development of Efficient Photocatalyst MIL-68(Ga)_NH_2_ Metal-Organic Framework for the Removal of Cr(VI) and Cr(VI)/RhB from Wastewater under Visible Light

**DOI:** 10.3390/ma15113761

**Published:** 2022-05-24

**Authors:** Lei Wu, Doudou Qin, Fan Fang, Weifeng Wang, Wenying Zhao

**Affiliations:** 1Institute of Polymer Materials, School of Materials Science and Engineering, Chang’an University, Xi’an 710064, China; q11044011@163.com (D.Q.); 2020231089@chd.edu.cn (W.W.); zey0916@126.com (W.Z.); 2Xi’an Thermal Power Research Institute Co., Ltd., Xi’an 710043, China

**Keywords:** MIL-68(Ga)_NH_2_, photocatalyst, Cr(VI), RhB, wastewater treatment

## Abstract

Severe environmental pollution is caused by the massive discharge of complex industrial wastewater. The photocatalytic technology has been proved as an effective way to solve the problem, while an efficient photocatalyst is the most critical factor. Herein, a new photocatalyst MIL-68(Ga)_NH_2_ was obtained by hydrothermal synthesis and were characterized by PXRD, FTIR, ^1^H NMR, and TGA systematically. The result demonstrates that MIL-68(Ga)_NH_2_ crystallized in orthorhombic system and *Cmcm* space group with the unit cell parameters: *a* = 36.699 Å, *b* = 21.223 Å, *c* = 6.75 Å, *V* = 5257.6 Å^3^, which sheds light on the maintenance of the crystal structure of the prototype material after amino modification. The conversion of Cr(VI) and binary pollutant Cr(VI)/RhB in wastewater under visible light stimulation was characterized by the UV-vis DRS. Complementary experimental results indicate that MIL-68(Ga)_NH_2_ exhibits remarkable photocatalytic activity for Cr(VI) and the degradation rate reaches as high as 98.5% when pH = 2 and ethanol as hole-trapping agent under visible light irradiation with good reusability and stability. Owing to the synergistic effect between Cr(VI) and RhB in the binary pollutant system, MIL-68(Ga)_NH_2_ exhibits excellent catalytic activity for both the pollutants, the degradation efficiency of Cr(VI) and RhB was up to 95.7% and 94.6% under visible light irradiation for 120 min, respectively. The possible removal mechanism of Cr(VI)/RhB based on MIL-68(Ga)_NH_2_ was explored. In addition, Ga-based MOF was applied in the field of photocatalytic treatment of wastewater for the first time, which broadened the application of MOF materials in the field of photocatalysis.

## 1. Introduction

With growing industrialization and urbanization, the composition of industrial wastewater is becoming more and more complicated, usually containing a lot of heavy metal ions, such as Cr(VI), Pb(II), and organic dyes such as RhB, MG, and MB, which have high toxicity, teratogenicity, and carcinogenicity, and resulting in a large amount of emission causing great threat to the ecological system and human health [1,2,3,4]. It appears thereby urgent to find an effective way to treat industrial wastewater with complex systems. At present, the main processes aiming to enable this rely on three different types of ways [5]: physical adsorption, chemical precipitation, and reduction method. However, most of these traditional treatment methods are only for a single pollutant, and always with high cost and low efficiency. Owing to the infinite availability and sustainability of the solar energy, the photocatalytic technology has become an efficient and eco-friendly approach to deal with wastewater while avoiding the secondary pollution. The key of this technology lies in the choice of photocatalyst. At present, the photocatalysts used for water purification of multiple pollutants mainly focus on inorganic semiconductor photocatalysts (e.g., TiO_2_, ZnO, Fe_2_O_3_) [6,7,8,9,10]. Although these systems have beneficial features, the application of traditional photocatalyst is limited, because of the restricted utilization of solar energy, lower specific surface area, low electron-hole separation efficiency, and limited function adjustability. Therefore, it remains a challenge to search or design novel, efficient, and tailored functional photocatalyst.

Metal-organic frameworks (MOFs), composed of metal nodes/metal-containing clusters coordinated by organic ligands, are a new kind of porous material. Owning to the merits of their high specific surface areas, adjustable porosity, unsaturated metal centers, MOFs possess huge application potential in gas storage, separation, sensing, linear optical properties, and catalysis [11,12,13,14,15]. In 2006, Mahata and co-workers first reported the degradation of different dyes with MOFs under the irradiation of mercury vapor lamp [16]. Since then, numerous studies involving MOFs as photocatalyst have been reported [17,18,19,20]. Because most of MOFs exhibit a wide light response range with the edge in the range of semiconductor band gap, MOFs are generally regarded as a class of semiconductor [21]. However, different from the traditional semiconductor, the band gap of MOFs can be adjusted flexibly by changing the organic component or inorganic component [22]. This advantage provides more selectivity for MOFs as photocatalyst. Amounts of exploration have been undertaken to expand the potential application of MOFs in the field of photocatalysis, and various reviews have been published to predict a bright future of MOFs as efficient and sustainable photocatalysts [17,23,24,25].

The structural stability and reusability of the photocatalyst are critical for photocatalytic reaction. Compared to divalent transition metals (such as Zn^2+^, Cd^2+^, or Co^2+^), high valence metals (such as Cr^3+^, Fe^3+^, or Zr^4+^) usually endow MOFs excellent thermal, chemical, and water stability. Based on high stability and large specific surface area, MIL-series materials have attracted much attention and made a lot of achievements as photocatalysts for wastewater treatment [26,27,28,29,30,31]. Du et al. successfully prepared a series of MIL-53(M) (M = Al, Fe, Cr) for the degradation of organic dye MB and found that MIL-53(Fe) had a low catalytic activity, and degradation efficiency of MB was only 3% under visible light irradiation for 40 min [26]. Jing et al. proved that MIL-68(Fe) exhibits high catalytic efficiency in Cr(VI) or mixed pollution Cr(VI)/MG, with Cr(VI) and MG degradation rates of 94.5% and 80.3%, respectively, under visible light irradiation for 240 min [27]. Shen et al. reported MIL-125(Ti)_NH_2_ with good photocatalytic activity for Cr(VI) of 91% under visible light irradiation [29]. Liang et al. highlighted that MIL-68(In)_NH_2_ reveals excellent photocatalytic conversion rate for Cr(VI) [30]. Later, Pi et al. introduced multi-functional multiwalled carbon nanotube (MWCNT) into MIL-68(In)_NH_2_ and promoted the conversion efficiency for the Cr(VI) [31]. So far, most research about MOFs photocatalysts focus on the degradation for unitary pollutant system, while the studies for multiple pollutants system are quite deficient or the degradation efficiency of the MOFs photocatalysts is undesirable.

MIL-68-type materials, reported by Férey and coworkers, are prepared from trivalent metal cations of d- and p-blocks (such as V^3+^, Fe^3+^, Al^3+^, In^3+^, or Ga^3+^) and terephthalic acid under solvothermal synthesis conditions [32,33,34,35,36]. Compared to the other isostructural materials, MIL-68(Ga), despite both higher apparent surface areas and outstanding thermal stability [36], has been surprisingly much less investigated. Indeed, as far as we know, the research of the photocatalytic performance based on Ga-based MOFs for photocatalytic wastewater treatment has not been reported so far. Thus, MIL-68(Ga) was selected as the initial material in our research. Through preliminary research, it was discovered that MIL-68(Ga) is non-responsive to visible light, which drove us to look for the ways to optimize its photocatalytic activity. Inspired by previous studies about adjusting the optical properties of MOFs by introducing specific substituents, the functionalization with the amino group of MIL-68(Ga) became our preferred strategy [37,38]. In the amino group, there exists a lone pair of electrons that can interact with the *π**-orbitals of the benzene ring to yield an elevated visible light response [17]. Moreover, the introduction of polar amino group into the MIL-68(Ga) may enhance the interaction between adsorbent and adsorbate, which is conducive to photocatalytic performance. Herein, MIL-68(Ga)_NH_2_ was synthesized by means of pre-functionalized modification with 2-aminoterephthalic acid as organic ligands to broaden visible light response range. In addition, the application of MIL-68(Ga)_NH_2_ as photocatalyst in wastewater purification of different pollution systems (Cr(VI) and Cr(VI)/RhB) was investigated and reported for the first time, which indicated excellent photocatalytic activity under visible light irradiation.

## 2. Experimental Section

### 2.1. Reagents

Gallium nitrate hydrate (Ga(NO_3_)_3_·*x*H_2_O), terephthalic acid (H_2_BDC), hydrofluoric acid (HF), Rhodamine B (RhB), and 2-aminoterephthalic acid (H_2_BDC-NH_2_) were purchased from Aladdin Reagent Co., Ltd. (Shanghai, China) Potassium dichromate (K_2_Cr_2_O_7_) was from Sinopharm Chemical Reagent Co., Ltd. (Shanghai, China) N, N-dimethylformamide (DMF) and anhydrous methanol were obtained by Aldrich. All materials were of analytical grade and used directly. For the calculation of molar quantities of the gallium source, the metal precursor has been considered as anhydrous.

### 2.2. Synthesis of MIL-68(Ga)_NH_2_

Ga(NO_3_)_3_·*x*H_2_O (0.491 g, 1.92 mmol) and H_2_BDC-NH_2_ (0.1165 g, 0.64 mmol) were mixed in DMF (6.18 mL), followed by HF (840 μL, 0.38 mmol). The uniformly mixed reactants were transferred to a 25 mL reaction kettle equipped with Teflon-liner, and heated at 125 °C for 5 h in a pre-heated oven. After cooling to room temperature, light-yellow crude product MIL-68(Ga)_NH_2_ was filtered and washed by DMF with the yield of around 59%. Finally, the appropriate activation method was adopted to remove the H_2_O and DMF in the pores. The crude samples were immersed in anhydrous methanol for 72 h and the solvent was replaced with fresh anhydrous methanol three times a day. Then the soaked products were filtered and heated in a vacuum oven at 200 °C for 10 h to get activated MIL-68(Ga)_NH_2_. The yield was about 75%.

### 2.3. Synthesis of MIL-68(Ga)

MIL-68(Ga) was produced referring to a previously reported formula [36]. H_2_BDC (0.200 g, 1.20 mmol), Ga(NO_3_)_3_·*x*H_2_O (0.415 g, 1.60 mmol), and DMF (10.00 mL) were mixed uniformly. Then the mixture was shifted to a 25 mL reactor equipped with Teflon-liner, and heated at 100 °C for 10 h. After cooling to room temperature, white crude sample MIL-68(Ga) was filtered and washed by DMF. The yield was about 75%. The activated MIL-68(Ga) sample was obtained according to ref [36].

### 2.4. Measurements

The scanning electron microscopy (SEM) photographs were taken by a Philips XL 30 FEG microscope. The ^1^H NMR spectra were performed on a Bruker 400 UltraShield^TM^ with tetramethylsilane as the standard. The powder XRD(PXRD) patterns were carried out on a STOE STADI-P diffractometer equipped with Cu K*_α_*_1_ radiation (*α* = 1.5406 Å). The FTIR spectra were performed on a BRUKER TENSOR II spectrometer by the means of KBr pellets. The thermogravimetric (TG) analysis was tested on a TA DSC/TGA Discovery SDT 650 simultaneous thermal analyzer at the heating rate of 5 °C min^−1^ from room temperature to 800 °C in N_2_ atmosphere. Nitrogen adsorption isotherms were recorded on a Micromeritics Tristar II 3020 apparatus. The specific surface area was estimated according to the Brunauer–Emmett–Teller (BET) and Langmuir models [39,40]. The UV-vis diffuse reflectance spectra (UV-vis DRS) was recorded on a UV-vis spectrophotometer TU-1950. The samples were scanned in the region from 800 nm to 230 nm with the white BaSO_4_ as the blank group. The Mott–Schottky measurement was tested on a CHI760D workstation. The study was conducted with a three electrodes system under four intermittent visible light exposures. The photocurrent measurement was carried out on a CHI1030B workstation. The sample was scanned at different frequencies (0.5 kHz, 1.0 kHz, 1.5 kHz) with a scanning voltage range of −0.6~0.8 V. A 300 W Xe lamp (Beijing Education Jinyuan Technology Co., Ltd., Beijing, China) with a 420 nm cut-off filter was used as a visible light source.

### 2.5. Photocatalytic Degradation Experiments

In order to analyze the photocatalytic activity of MIL-68(Ga)_NH_2_ on Cr(VI), Cr_2_O_7_^2−^ was used to simulate Cr(VI) contaminant in sewage. The degradation experiment of Cr(VI) was performed in a 250 mL three-port quartz reactor at room temperature and N_2_ atmosphere under visible light irradiation. The distance between the light source and the solution level was fixed at 13 cm. A total of 50 mg of MIL-68(Ga)_NH_2_ and different doses of hole-trapping agent were mixed into 50 mL of Cr(VI) aqueous solution (20 ppm). An appropriate amount of H_2_SO_4_ (0.1 M) or NaOH (0.1 M) was added to regulate the pH values of reaction system. To achieve adsorption–desorption balance, the system was kept stirring for 1 h in the dark. Then, Xe lamp was turned on for visible light irradiation and about 2 mL of suspension was withdrawn from the beaker per 20 min. Finally, the supernatant was centrifuged and collected for measuring with a UV-vis spectrophotometer with a diphenylcarbazide method [39].

The study of the photocatalytic activity of MIL-68(Ga)_NH_2_ for dual pollutants was similar to the aforementioned method, but the mixture of 25 mL of Cr_2_O_7_^2−^ (20 ppm) and 25 mL of RhB (30 ppm) was used to simulate complex industrial wastewater instead of Cr_2_O_7_^2−^ solution. About 4 mL of suspension was withdrawn from the beaker per 20 min for testing the concentration of two pollutants (Cr(VI) and RhB), with 2 mL for each one. The concentrations of RhB was measured with the TU-1950 UV-vis spectrophotometer, and the concentrations of Cr(VI) was measured with a diphenylcarbazide method.

## 3. Results and Discussion

Figure 1a displays the crystal morphology of the MIL-68(Ga)_NH_2_, which showed homogeneous hexagonal lumps and the particle size of approximately 3 × 5 μm.

Because the size of the single crystal was inappropriate for performing SCXRD, the unit-cell parameters of MIL-68(Ga)_NH_2_ was determined by the approach of PXRD with DICVOL4 algorithm [41,42]. The result demonstrated that MIL-68(Ga)_NH_2_ crystallized in orthorhombic system and *Cmcm* space group. The unit cell parameters were as follows: *a* = 36.699 Å, *b* = 21.223 Å, *c* = 6.75 Å, *V* = 5257.6 Å^3^, and Figure of Merit F(30) = 41.6. The lattice parameters of the two isomorphs are compared as shown in Table 1. The comparison revealed that the two structures have little variation. In addition, the characteristic peaks in the PXRD pattern of MIL-68(Ga)_NH_2_ well matched with the simulated one of MIL-68(Ga) without significant difference (Figure 1b). Therefore, the results indicated that the insert of amino functional groups did not change the crystal structure of prototype MIL-68(Ga) and the Kagomé topological structure was sustained. MIL-68(Ga)_NH_2_ was built up with gallium-oxygen octahedral units linked by μ-OH and the terephthalate ligands to establish two kinds of the channels in the skeleton.

The FTIR spectra of H_2_BDC-NH_2_, MIL-68(Ga), and MIL-68(Ga)_NH_2_ were investigated as shown in Appendix A. Compared to the FTIR spectrum of MIL-68(Ga), two vibration peaks at 1626 and 1258 cm^−1^ clearly appeared in the spectrum of MIL-68(Ga)_NH_2_, which are assignable to the N-H bending vibration and C-N stretching of aromatic aminos, respectively [43]. Meanwhile, the two weak peaks at 3381 and 3483 cm^−1^ can be observed, which are considered as symmetric and asymmetric stretching vibrations of the aromatic aminos [44]. Because these two peaks are overlapped by a broad peak of H_2_O and the bridged hydroxyl of the skeleton, the intensity is much less than the ones in the FTIR spectrum of H_2_BDC-NH_2_. The results demonstrated that the amino functional groups have been successfully introduced into MIL-68(Ga) without coordination.

Appendix A displays the ^1^H NMR spectra of H_2_BDC-NH_2_, as-synthesized MIL-68(Ga)_NH_2_ and activated MIL-68(Ga)_NH_2_. The three independent sets of signals at around *δ* = 7.87 ppm, 7.66 ppm, 7.36 ppm, which are assigned to the phenyl protons of the organic ligand, can be found in all the spectra. It confirms the existence of the terephthalate ligand in the framework. Moreover, the three additional signals of the DMF molecules marked with asterisk can be clearly found in the ^1^H NMR spectrum of the as-synthesized MIL-68(Ga)_NH_2_ (Appendix A). The corresponding signals disappeared in the spectrum of the activated sample (Appendix A), which indicates no existence of guest DMF molecules in the activated samples. Therefore, the formula of as-synthesized MIL-68(Ga)_NH_2_ can be derived by integrating these sets of signals, which is proposed as Ga(BDC-NH_2_)(OH)·(DMF)*_x_*(H_2_O)*_y_* (*x* ≈ 1.2).

The TG curve of MIL-68(Ga)_NH_2_ sample is shown in Appendix A. There are three obvious weight losses. The first weight loss occurs from 30 °C to 90 °C, which can be considered as the loss of trapped water (obs.: 7.3%; calc.: 7.1%). The second weight loss, between 90 °C and 315 °C, corresponds as the removal of residual DMF in the pores (obs.: 24.6%; calc.: 24.8%). Finally, the framework structure collapses from 350 °C, which indicates that MIL-68(Ga)_NH_2_ has excellent thermal stability. In addition, combined with the analysis results of ^1^H NMR, the derivation formula of MIL-68(Ga)_NH_2_ is considered to be Ga(OH) (BDC-NH_2_)·(DMF)*_x_*(H_2_O)*_y_* (*x* ≈ 1.2 and *y* ≈ 1.5).

The N_2_ adsorption behavior of MIL-68(Ga)_NH_2_ activated in the aforementioned condition is depicted in Figure 2. The N_2_ sorption isotherm showed a dramatic absorption behavior in the low-pressure region (10^−5^ to 10^−1^ atm), which is the characteristics of the I-type isotherm of microporous solids. Determined in the 0.00433–0.04950 *p*/*p*^0^ range, the BET and Langmuir surface areas of MIL-68(Ga)_NH_2_ reached 790 and 840 m^2^ g^−1^ (micropore volume: 0.33 cm^3^ g^−1^), respectively. However, the corresponding values of MIL-68(Ga) were 1117 and 1410 m^2^ g^−1^ (micropore volume: 0.46 cm^3^ g^−1^) reported by Férey and coworkers [36]. It was found that the BET and Langmuir specific surface areas of MIL-68(Ga)_NH_2_ lost 29% and 40%, respectively, and the pore volume lost about 28%, which may be considered as the partial occupation of pore by the insert of amino groups. Many instances displayed that substituting groups have great influence on the pore volume [45,46,47]. Nevertheless, MIL-68(Ga)_NH_2_ possesses a considerably high porosity.

The room temperature UV-vis DRS of MIL-68(Ga)_NH_2_ and MIL-68(Ga) samples as well as the H_2_BDC-NH_2_ ligand are illustrated in Figure 3. The reflectance was converted to absorbance by the Kubelka–Munk method. The primary optical absorption band around 365, 390 nm for MIL-68(Ga)_NH_2_ and H_2_BDC-NH_2_, respectively, should be arisen from the *n*–*π** transition of the lone pair electrons of -NH_2_ in the ligand [17,48]. Owing to the perturbation of the transition metal, the blue shift of optical response region is observed for MIL-68(Ga)_NH_2_ versus free ligand [29]. Meanwhile, the optical response edge of the MIL-68(Ga)_NH_2_ sample is evidently shifted to longer wavelength compared to the prototype MIL-68(Ga). The main absorption edges of MIL-68(Ga)_NH_2_ and MIL-68(Ga) samples are 450 and 320 nm, respectively, corresponding to the band gaps (Eg) of 2.76 and 3.88 eV (Eg = 1240/wavelength). It reveals that both of the samples present semiconductor properties. The comparison of the two main absorption peaks shows that modification of the amino group is an effective means to expanding photoabsorption edge of MOFs into the visible light region, which is reflected obviously in color changing of the samples as illustrated in the inset of Figure 3.

The photochemical properties of the samples have been explored (Figure 4a). When MIL-68(Ga)_NH_2_ is an electrode under intermittent illumination of visible light, it showed an apparent photocurrent response. When the light was turned on, the current density rapidly increased and gradually tended to saturation after a period of time. When the excitation light source was turned off, the current returned to the initial state due to the recombination of the photoelectrons and holes. Instead, MIL-68(Ga) showed almost no photocurrent response during the test procedure. This phenomenon indicated that MIL-68(Ga)_NH_2_ could generate photoelectrons and holes under the stimulation of visible light. In addition, the Mott–Schottky was measured in darkness to further explore the electrochemical properties of MIL-68(Ga)_NH_2_ (Figure 4b). The slope of C^−2^-E is positive which confirms MIL-68(Ga)_NH_2_ was an *n*-type semiconductor [29]. From the *x*-intercept of the oblique line, the reduction potential of MIL-68(Ga)_NH_2_ was about ca. −0.9 V vs. Ag/AgCl pH = 6.8, estimated to be about −0.7 V vs. NHE pH = 6.8, which was much lower than the value of Cr(VI)/Cr(III) (+0.51 V, pH = 6.8) [49]. This indicates that it is thermodynamically possible for MIL-68(Ga)_NH_2_ to reduce Cr(VI) to Cr(III).

The photocatalytic properties of MIL-68(Ga)_NH_2_ were studied under visible light illumination with Cr(VI) solution in the form of Cr_2_O_7_^2−^ as a simulated pollutant. As displayed in Figure 5a, almost no removal of Cr(VI) without light stimulation or catalyst was seen, which illustrated that MIL-68(Ga)_NH_2_ was a photocatalyst. Compared with commercial TiO_2_ (P25), MIL-68(Ga)_NH_2_ showed higher catalytic activity, which suggested MIL-68(Ga)_NH_2_ was an efficient catalyst in response to visible light. As displayed in Figure 5b, the conversion rate of Cr(VI) in the presence of MIL-68(Ga)_NH_2_ without hole trapping agent was only 53% after 180 min of visible light stimulus. However, the conversion rate increased in varying degrees with the addition of different types of hole trapping agent, which confirmed that MIL-68(Ga)_NH_2_ was an electronic photocatalyst. Amongst different hole trapping agents, ethanol exhibited the highest effect as a hole trapping agent. When 200 µL ethanol hole trapping agent was added, the conversion rate of Cr(VI) reached up to 95.9% after 120 min of visible light irradiation. The reason may be that the ethanol hole trapping agent is easier to be adsorbed on the surface of the catalyst to form reductive free radicals, which react with the hole generated by the catalyst to effectively separate the electron-hole pairs [31]. As the amount of ethanol increased (Figure 5c), the removal rate of Cr(VI) speeded up, accompanied by the maximum rate of 98.5% with 300 µL ethanol consumption. However, the degradation rate gradually decreased when ethanol consumption reached 400 µL, which may be owing to too much ethanol trapping agent adsorbed on the surface of catalyst covering the reactive site.

The pH value of the system showed a significant effect on removal efficiency of Cr(VI). As shown in Figure 5d, with pH = 6, the adsorption efficiency of the catalyst in the dark reaction was observed as only 7% and the conversion rate of Cr(VI) was only 67% under the stimulus of visible light for 180 min. As the pH value decreased from 6 to 2, the adsorption efficiency in the dark reaction and the removal rate of Cr(VI) in the light reaction increased gradually. Accompanied by the degressive pH value, the surface zeta potential of MIL-68(Ga)_NH_2_ increased, which caused the surface of MIL-68(Ga)_NH_2_ to be more electropositive. It was conducive to adsorption of Cr_2_O_7_^2^^−^, which greatly improved the adsorption capacity and catalytic efficiency [50]. When pH = 2, the adsorption capacity of catalytic reached 25%, the degradation rate of Cr(VI) was up to 98.5% under visible light irradiation for 120 min.

The reusability and stability are important performance evaluation indexes of the photocatalyst. Four cycles of photocatalytic degradation of Cr(VI) were tested and the crystal structures of MIL-68(Ga)_NH_2_ before and after Cr(VI) reduction were characterized by PXRD. After each cyclic test, the catalyst was centrifugal separation and washed, and then added to the equal-concentration Cr(VI) aqueous. Because the amount of catalyst used in the cycle experiment was too small to meet the requirements of PXRD, the amount of catalyst and Cr(VI) aqueous was proportionally increased during the actual operation process. Meanwhile, the irradiation time was prolonged as well. The results of the four cycles are depicted in Figure 6a. The conversion rate of Cr(VI) remained at about 98.5% under visible light irradiation for 180 min. PXRD results (Figure 6b) confirmed that the crystal structure of the MIL-68(Ga)_NH_2_ was almost intact before and after the cycle experiment. This proved the good structural stability and recyclability of MIL-68(Ga)_NH_2_. Slight loss of activity may be attribute to the loss of catalyst during centrifugation and washing.

In the REDOX process of Cr(VI) to Cr(III), Cr(VI) capture photogenerated electrons as oxidant. Ethanol as the hole-trap-agent can promote electron-hole separation, so as to improve the photocatalytic efficiency of Cr(VI). However, ethanol as an organic solvent is less environmentally friendly. Therefore, it is necessary to find a better way to promote the separation of electrons and holes instead of the addition of ethanol. RhB is a typical organic dye pollutant in industrial wastewater. Many studies have shown that RhB can be oxidized and decomposed into water and CO_2_ by hole generated by photocatalysis. Inspired by this, RhB was introduced into the system to replace ethanol as a hole trapping agent. The degradation efficiency of unitary pollutant and dual pollutants are shown in Figure 7. In the dual pollutants system, Cr(VI) and RhB captures electrons and holes during the photocatalytic reaction process, respectively. The synergistic effect between the two pollutants leads to the rapid separation of photocarriers, which makes the photocatalytic degradation efficiency of RhB and Cr(VI) significantly higher than that of the unitary pollutant system. When pH = 2, MIL-68(Ga)_NH_2_ showed excellent activity on Cr(VI), and the degradation efficiency was as high as 95.7% under visible light irradiation for 2 h, which held twice efficiency compare with the system without RhB. Although it is not as effective as ethanol as trapping agent, this system is much more environmentally friendly. Not only does not introduce organic matter, but also degrade the organic dye pollutant RhB with the considerable efficiency of 94.6% as well as the efficient degradation of Cr(VI). The degradation efficiency of dual pollutants is higher than the reported porous inorganic composites g-C_3_N_4_/Na-bentonite, ZnFe_2_O_4_/Na-bentonite, ZnTiO_3_/Zn_2_Ti_3_O_8_/ZnO, TiO_2_, RP_0.01_TiO_2_, g-C_3_N_4_/TiO_2_ nanorods [51,52,53,54].

Furthermore, the mechanism of MIL-68(Ga)_NH_2_ photocatalytic treatment of sewage containing Cr(VI) and RhB was explored as shown in Figure 8. The gallium-oxygen inorganic clusters of MIL-68(Ga)_NH_2_ framework act as quantum dots connected by 2-amino-terephthalic acid and these organic ligands can perform as a light-absorbing antennae effectively transferring energy to the inorganic clusters [32,50]. When visible light irradiates MIL-68(Ga)_NH_2_, the electrons in the conduction band are excited to jump to the valence band, generating electrons and holes that have strong REDOX performance. The photogenerated electrons (e^−^) are transferred and react with the adsorbed Cr_2_O_7_^2−^ to form Cr(III), while h^+^ reacts with H_2_O in the system to form hydroxyl radical (·OH), which reacts with RhB to form H_2_O and CO_2_ and intermediates. Meanwhile, the photocatalytic activity of MIL-68(Ga)_NH_2_ is also consistent with the rapid separation of electron and hole. The large specific surface area of MIL-68(Ga)_NH_2_ and the existence of polar -NH_2_ functional groups are conducive to improving the interaction between adsorbents and catalysts, so that the electrons can transfer rapidly, greatly improving the photocatalytic efficiency. The possible reaction mechanism equations are as follows:**MIL-68(Ga)_NH_2_ + hv        MIL-68(Ga)_NH_2_ (h^+^ + e^−^)**(1)
**14H^+^ + Cr_2_O_7_^2−^ + 6e^−^            2Cr_3_^+^ + 7H_2_O**(2)
**h^+^ + H_2_O         • OH + H^+^**(3)
**• OH + RhB        CO_2_ + H_2_O + Intermediates**(4)

## 4. Conclusions

In conclusion, amino modified MIL-68(Ga) was successfully synthesized. The structure of prototype MIL-68(Ga) has been sustained after amino modified, despite the slight loss of apparent surface areas. Notably, UV-vis DRS shows that MIL-68(Ga)_NH_2_ has obvious visible light response. When pH = 2 and ethanol as hole-trapping agent, MIL-68(Ga)_NH_2_ exhibits remarkable photocatalytic activity and the degradation rate of Cr(VI) is as high as 98.5% under visible light irradiation holding outstanding stability and recyclability, which is far beyond the degradation efficiency of commercial TiO_2_(P25) for Cr(VI) under the same conditions. Meanwhile, MIL-68(Ga)_NH_2_ can also effectively degrade RhB and Cr(VI) simultaneously due to the existence of the synergistic effect between heavy metals and dyes. When pH = 2, the concentration ratio of Cr(VI) and RhB was 1:1.5, the degradation efficiency of Cr(VI) and RhB is up to 95.7% and 94.6%, respectively, under visible light irradiation for 120 min. In this work, Ga-based MOFs were applied in the field of photocatalytic treatment of wastewater for the first time and can be utilized as promising photocatalysts for the environmental cleanup and show great application prospect. It is expected that this work could expand the exploration, the utilization, and the property-control of MOFs as photocatalyst.

## Figures and Tables

**Figure 1 materials-15-03761-f001:**
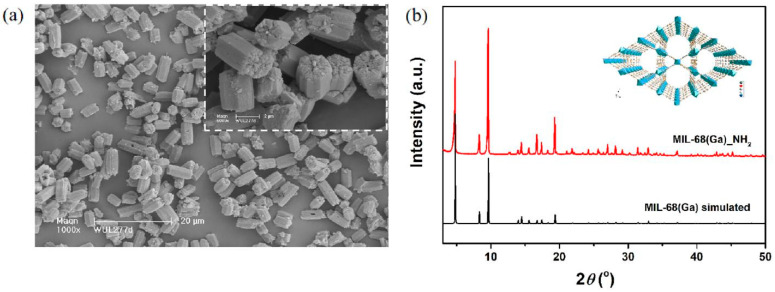
(**a**) SEM photograph of the as-synthesized MIL-68(Ga)_NH_2_; the inset is SEM photograph at high resolution. (**b**) Simulated PXRD pattern of MIL-68(Ga), black; PXRD pattern of as-synthesized sample of MIL-68(Ga)_NH_2_, red.

**Figure 2 materials-15-03761-f002:**
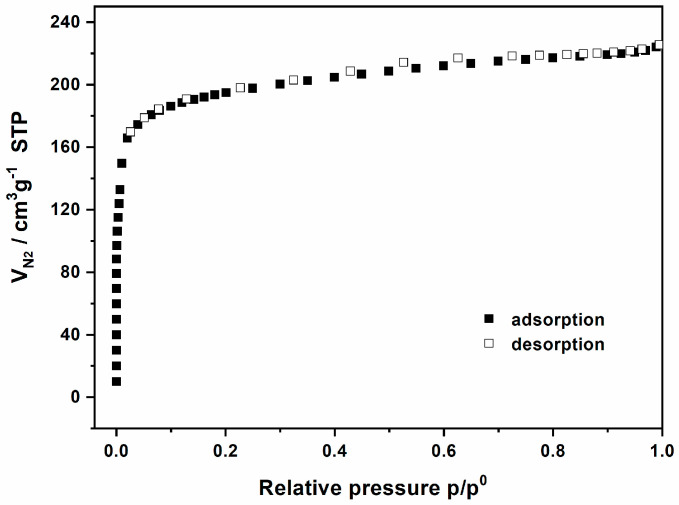
N_2_ sorption isotherms of MIL-68(Ga)_NH_2_. (Solid symbols, adsorption; open symbols, desorption).

**Figure 3 materials-15-03761-f003:**
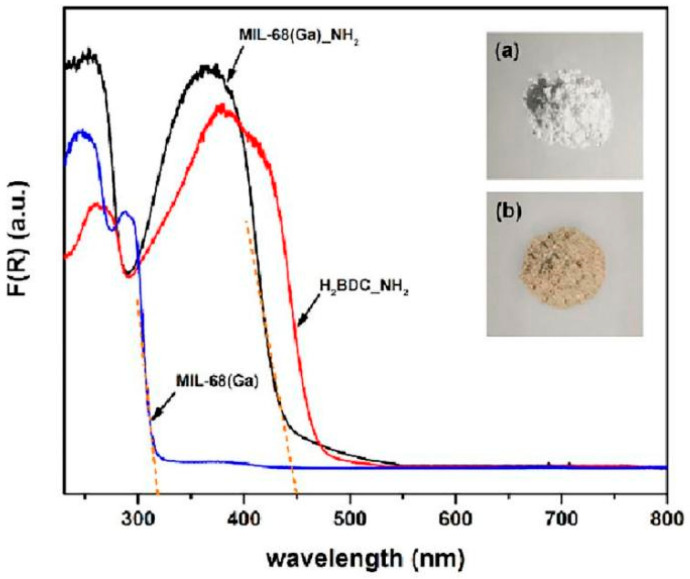
UV-vis DRS of MIL-68(Ga)_NH_2_ (black), MIL-68(Ga) (blue) and H_2_BDC_NH_2_ (red). The insets are the photographs of (**a**) MIL-68(Ga) and (**b**) MIL-68(Ga)_NH_2_ samples.

**Figure 4 materials-15-03761-f004:**
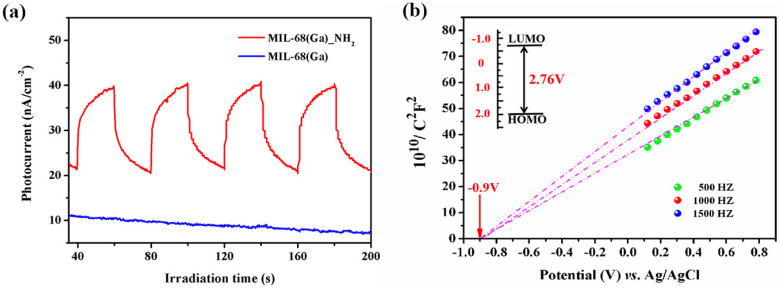
(**a**) Transient photocurrent response of MIL-68(Ga)_NH_2_ and MIL-68(Ga). (**b**) Typical Mott–Schottky plots of MIL-68(Ga)_NH_2_.

**Figure 5 materials-15-03761-f005:**
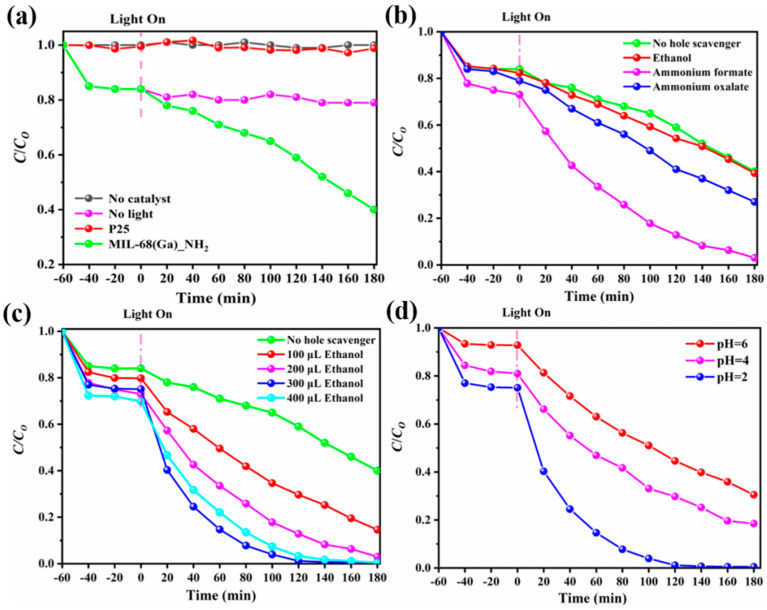
(**a**) Photocatalytic reduction of aqueous Cr(VI) under different conditions. (**b**) Photocatalytic activities of MIL-68(Ga)_NH_2_ for the reduction of aqueous Cr(VI) in the presence of various hole scavengers. (**c**) Photocatalytic activities of MIL-68(Ga)_NH_2_ for the reduction of aqueous Cr(VI) in the presence of different amount of ethanol hole trapping agent. (**d**) Photocatalytic activities of MIL-68(Ga)_NH_2_ for the reduction of aqueous Cr(VI) at different pH values.

**Figure 6 materials-15-03761-f006:**
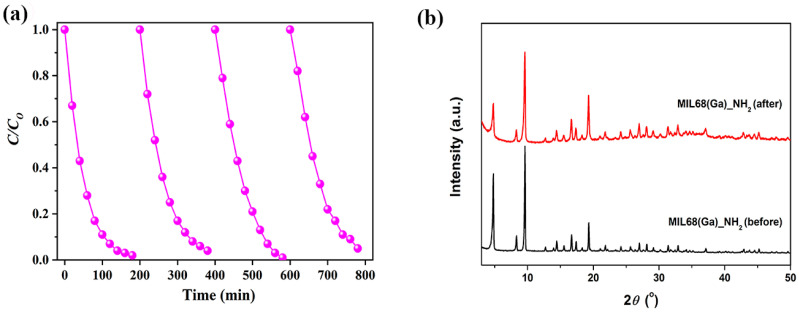
(**a**) Changes of the Cr(VI) concentration during the four repeated processes over MIL-68(Ga)_NH_2_. (**b**) PXRD patterns of MIL-68(Ga)_NH_2_ before (black) and after (red) reduction Cr(VI) for 180 min prolonged reaction.

**Figure 7 materials-15-03761-f007:**
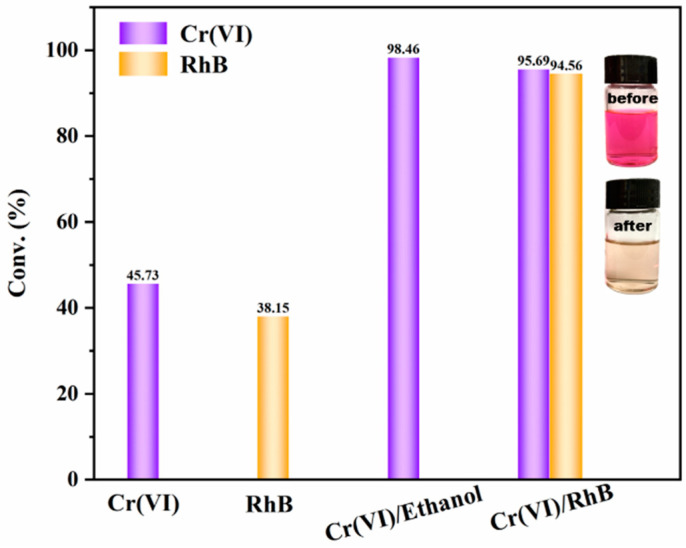
Effects of MIL-68(Ga)_NH_2_ on the degradation efficiency of pollutants in different systems.

**Figure 8 materials-15-03761-f008:**
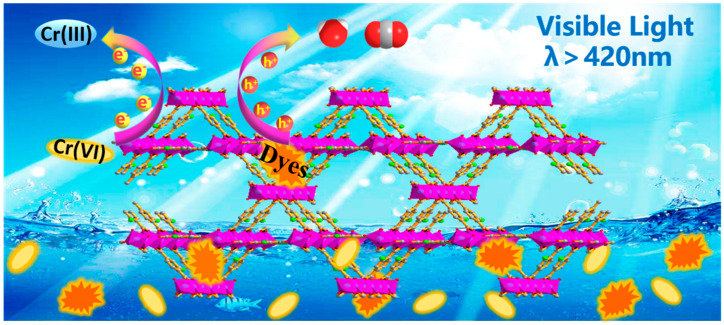
The schematic illustration of photocatalytic removal of Cr(VI) and RhB over MIL-68(Ga)_NH_2_ under visible light irradiation.

**Table 1 materials-15-03761-t001:** Unit-cell dimensions of MIL-68(Ga) [34] and MIL-68(Ga)_NH_2_. The inset is the structure of MIL-68(Ga).

Func.	*a* (Å)	*b* (Å)	*c* (Å)	*V* (Å^3^)	System	Space	F(30)
—	21.176	36.703	6.742	5240	Orthorhombic	*Cmcm*	—
–NH_2_	36.699	21.223	6.750	5257	Orthorhombic	*Cmcm*	41.6

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
