# Peer review of "Development of Efficient Photocatalyst MIL-68(Ga)_NH2 Metal-Organic Framework for the Removal of Cr(VI) and Cr(VI)/RhB from Wastewater under Visible Light"

_materials, 2022, doi:10.3390/ma15113761_

Round 1
Reviewer 1 Report
Dear Editor
The authors of the current work have presented an interesting work on wastewater treatment using a photocatalyst (MIL-68-NH2). This manuscript has shown new and reliable data regarding photocatalytic activity of MIL-68-NH2 toward Cr and Cr/RhB elimination and has been written in a good structure which makes it easy to understand. Nonetheless, there are still some minor issues that need to be addressed before considering publication. So, I would say Minor revision is required for the present work.
- The abstract needs to be strengthened by adding some quantitative data.
- Although it won't change the result, but it is suggested to simulate XRD pattern of MIL-68-NH2 and compare it with the experimental pattern in Fig.1.
- The SEM and XRD analyses of MIL-68 should be added as well.
- The authors have mentioned that "The main absorption edges of MIL-68(Ga)_NH2 and MIL-68(Ga) samples are 450 and 320 nm". How do the authors recognize those wavelengths i.e. 450 and 320 nm? It is better to draw tangent lines to show the exact location of those numbers. Moreover, why more accurate methods like the Tauc plot has not been utilized by authors?
- To study the recyclability of the photocatalyst, in addition to XRD, the FTIR should be investigated.
Author Response
Dear Reviewer:
We have answered your questions one by one and attached the reply in the file. Please check it. Thank you.

Reviewer 2 Report
see the file

Author Response

(The authors gave the same response as above.)

Reviewer 3 Report
The work presents a study on the development of an efficient photocatalyst for the removal of Cr(VI) and Cr(VI)/RhB from wastewater under visible light. The work is publishable after addressing some points:
- The English language must be improved throughout the manuscript.
- Please improve the introduction. It’s too branchy.
- Please see if the title can be improved such as: replacement of “Fabrication” with “Development , english check: “remove” replaced by “removal”.
- In the EXPERIMENTAL SECTION, Photocatalytic Degradation Experiments, it is unclear how much from the suspension is taken: 2mL or 4 mL?
- Please see il all figures are at the same size. As example figure 3 is too small to be seen.
Author Response

(The authors gave the same response as above.)

Reviewer 4 Report
. This journal is committed to engaging with a wider public in order to promote the potential benefits cutting-edge engineering research. Please describe in specific terms the potential impact of your work on the wider public. -
- Ideally, this article demonstrate how research results can be used in process engineering design and practice. Please outline briefly the engineering aspects in your paper (as opposed to scientific aspects).
-
Significance of the work. What is the significance of the substrate, process, methodology, results, etc.? Why is it interesting?
-
Contribution to the field. How does this paper advance the current knowledge?
- Ideally, this work related to pollution treatment must take an integrated approach to pollution control preventing transfer of pollution from one environment to another (e.g. from water to solid waste). To follows such an approach, elaborate how any treatment effluents, spent absorbents etc. can be treated or disposed safely, avoiding transfer of pollution to another environmental medium.
-
What are the bottlenecks of this work and how did you mitigate the impacts attributed to them?
- What are the technological innovations of the work?
- How did you do quality control (QC) and quality assurance (QA) on the obtained data to validate the conclusions?
- What is the operational cost of the use of the technology for this purpose (US$/meter cubic of treated effluent)?
-
This work was undertaken in laboratory scale. How to adjust and optimize the operational conditions for pilot (larger) scale?
- Make a table of comparison of this present work and other similar techniques from previously published study in terms of operational parameters and operational cost; afterwards, please give a critical analysis on its technical feasibility and applicability for upscaling this treatment process.
12. At industrial scale, what reactor configurations and technologies are used to refine the recovered product. Provide good discussion and examples.
13. What about thermal stability of the material obtained? 14. A comprehensive managerial insight should be provided in this paper. 15. Authors must do a sufficient literature survey in this area and many progress in this research topic is largely missed. Doi: 10.1016/j.jenvman.2020. 110839 Doi:10.1016/j.msec.2019.110420 - Pls ask a native English speaker from the MDPI's proofreading service to revise and proofread your revised work before re-submission and attach the certificate of their service with your revised work.
Please respond to all of those comments in the revised manuscript by pointing out precisely and concisely on which page and in which line you have incorporated your response one by one.
Author Response

(The authors gave the same response as above.)

Round 2
Reviewer 2 Report
accept
Author Response
We have communicated with the editorial department and there is no problem.
Reviewer 4 Report
The work has been revised. Nevertheless, there are minor mistakes in reference styles for {9-10]. Please correct them as follows before its acceptance.
Zhu, M., Kurniawan, T.A., Yanping, S., Othman, M.H.D., Avtar, R., Fu, D., Hwang, G.H.. Fabrication, characterization, and application of ternary magnetic recyclable Bi2WO6/ BiOI@Fe3O4 composite for photodegradation of tetracycline in aqueous solution. J. Environ. Manage. 2020, 270, 110839.
Zhu, M., Kurniawan, T.A., You Y., Othman, M.H.D., Avtar, R. 2D Graphene oxide (GO) doped p-n type BiOI/Bi2WO6 as a novel composite for photodegradation of bisphenol A (BPA) in aqueous solution under UV vis irradiation. Mater. Sci. Eng. C 2020, 108, 110420
